# Population Status of the Globally Threatened Long-Tailed Duck *Clangula hyemalis* in the Northeast European Tundra

Oleg Mineev *, Yurij Mineev, Sergey Kochanov and Alexander Novakovskiy 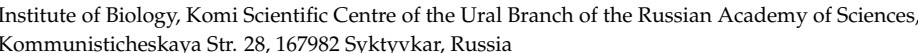

Institute of Biology, Komi Scientific Centre of the Ural Branch of the Russian Academy of Sciences, Kommunisticheskaya Str. 28, 167982 Syktyvkar, Russia

* Correspondence: mineev@ib.komisc.ru

**Abstract:** Arctic Russia is home to more than 90% of all Long-tailed Ducks in the *Clangula hyemalis* species from the Western Siberia/Northern Europe population. The breeding population in European Russia was estimated to be about 5 million birds in the 1960s, while today, estimates have declined to 1 million birds. Up until now, the main reasons for the overall population decline of the Long-tailed Duck were related to wintering conditions in the Baltic Sea. Our data indicate that the loss or deterioration of key breeding habitats in the Arctic regions of Russia is one important factor influencing the rapid population decline. Many key breeding habitats of the Long-tailed Duck were completely lost in the Bolshezemelskaya tundra, as this area was transformed into major oil and gas extraction sites. The transformation of these sites increased the disturbance and oil pollution of adjacent habitats, leading to the direct loss of certain key nesting sites and a marked and rapid decline of the breeding population of the Long-tailed Duck in the Bolshezemelskaya tundra. Oil-spills during transportation by sea may also be an important factor of decline in the Long-tailed Duck population. Meanwhile, in the Malozemelskaya tundra, which did not experience oil and gas development, the breeding population over the last decades remained stable. Urgent establishment of new protections in key breeding areas in Arctic Russia, sustainable population management, and new research programs are necessary for the conservation and enhancement of this globally threatened species.

**Keywords:** Long-tailed Duck; population status; Northeast European tundra

## 1. Introduction

The Long-tailed Duck breeds predominantly in Arctic tundra habitats, moving to marine areas for the non-breeding season [1]. The species has a high Arctic circumpolar breeding distribution, and within the Eurasian region, it breeds predominantly in Russia, with smaller populations in Finland, Sweden, Norway, Iceland, and Greenland. The Long-tailed Duck is a long-distance migrant. The majority of the birds belonging to the Northern European/Western Siberian population overwinter in the Baltic Sea [2].

In the 1990s, the total wintering population of the species in the Baltic Sea was estimated to be about 4.3 million birds [3], while the entire Western Siberia/Northern Europe population was estimated to be 4.6 million birds [2]. Results of large-scale surveys of wintering birds conducted in the Baltic Sea from 2007–2009 indicated a decline in the population to about 1.5 million birds [4], leading to an overall estimate of 1.6 million birds for the Western Siberia/Northern Europe population [5]. A marked decrease in numbers between the 1990s and the late 2000s was reported in several studies [6–8].

Up until now, the main reasons for the overall decline in population were related to the conditions of wintering ducks in the Baltic Sea. The Baltic Sea hosts about 90% of all wintering Long-tailed Ducks from the Western Siberia/Northern Europe population [2]. Recent studies using genetic markers [9] and geolocator tracking [10,11] confirmed that the Baltic Sea is the main wintering area for Long-tailed Ducks breeding in the tundra zone

of European Russia and Western Siberia. Two main threats to the population decline of the Long-tailed Duck were identified in the International Single Species Action Plan for the Conservation of the Long-tailed Duck: oil pollution from shipping and gillnet fishing bycatching [5]. In winter, Long-tailed Ducks form aggregated flocks, concentrating in specific areas of the Baltic Sea, where bivalves are abundant and accessible; therefore, they are highly vulnerable to oil pollution from shipping in such areas. It was estimated that anywhere from 50,000–100,000 Long-tailed Ducks were killed annually from oil spills in the central Baltic Sea from the 1990s to the early 2000s [12]. However, the number of confirmed oil spills in the Baltic Sea decreased from 472 in 2000 to 62 in 2018 [13]. The other major threat is the incidental catch of ducks during gillnet fishing, which kills about 90,000 birds in the Baltic Sea annually [14,15].

Low productivity and the increased mortality of adult birds from anthropogenic causes (such as oil discharges, fishing bycatching, and hunting in wintering areas) are also suggested as possible causes for the rapid population decline of the Long-tailed Duck since the mid-1990s [6]. It is probable that, similar to other sea ducks, the Long-tailed Duck is a relatively long-lived *K*-selected species, with a reasonably stable population, due to high adult survival and usually low numbers of ducklings. In such species, even small changes in the adult survival rate can affect population stability [16].

Due to the large decline in the number of ducks wintering in the Baltic Sea since the mid-1990s (equivalent to a 59% decline in the global population over three generations), the Long-tailed Duck was classified as 'Vulnerable' on the International Union for Conservation of Nature (IUCN) Red List in 2012 [5]. The majority of the European population of Long-tailed Ducks are found in northeast European Russia. The key breeding areas are located in Arctic freshwater habitats between the Kaninskij and Yugorskij Peninsulas [17,18].

The objective of this study is to assess changes in the recent distribution and abundance of the Long-tailed Duck within its core breeding range. Such data are critically important for a reliable assessment of the conservation status of this globally threatened species.

## 2. Materials and Methods

Field surveys of breeding Long-tailed Ducks were conducted from 1973–2022. These surveys covered a major part of the northeast European tundra in Russia, located in the Nenets Autonomous District and in the Komi Republic between the Kaninskij Peninsula and Yugorskij Peninsula (43°13′14″ E–65°03′36″ E) (Figure 1).

Geographical survey areas were divided into five major regions: the Kaninskaya tundra, the Timanskaya tundra, the Malozemelskaya tundra (hereafter the MZT), the Bolshezemelskaya tundra (hereafter the BZT), and the Yugorskij Peninsula [19]. The Pechora River delta, as an interzonal element, was not included in our study because it does not host typical habitats for the species. Long-term field surveys of the Long-tailed Duck were conducted in the most important breeding areas in the MZT, located west of the delta (1977–1979, 1982, 1986–1996, 1998, 2000–2005, 2007–2010, 2018, 2019, 2020, 2022); in the BZT, located east of the delta (1973–1980, 1986, 1989, 1992, 1997, 1999, 2001, 2003, 2006–2009, 2011–2017); and on the Yugorskij Peninsula (1981–1984, 1987, 2010, 2012, 2015). In other regions, detailed surveys were performed only in certain years, mainly during aircraft routes.

Land-based and boat surveys were used. The total length of the land-based surveys was 7670 km. Standard line-transect methods to determine the breeding densities of ducks (counting ducks along selected routes of up to 5 km, within 500-meter-wide zones) were applied [20,21]. Nesting pair abundance was estimated by counting females with or without males and females with broods during the local breeding period. A nest search was also performed. The total length of the boat surveys was 9300 km. Breeding densities of Long-tailed Ducks were estimated per 10 km of the boat route. For the population dynamics analysis of the Long-tailed Duck, we used data obtained from the BZT and the MZT, as they were enough to make the analysis representative. Data obtained from the Yugorskij

Peninsula were not enough (only 8 years of survey) for the representative analysis. We used these data to illustrate the population density of the Long-tailed Ducks in the region.

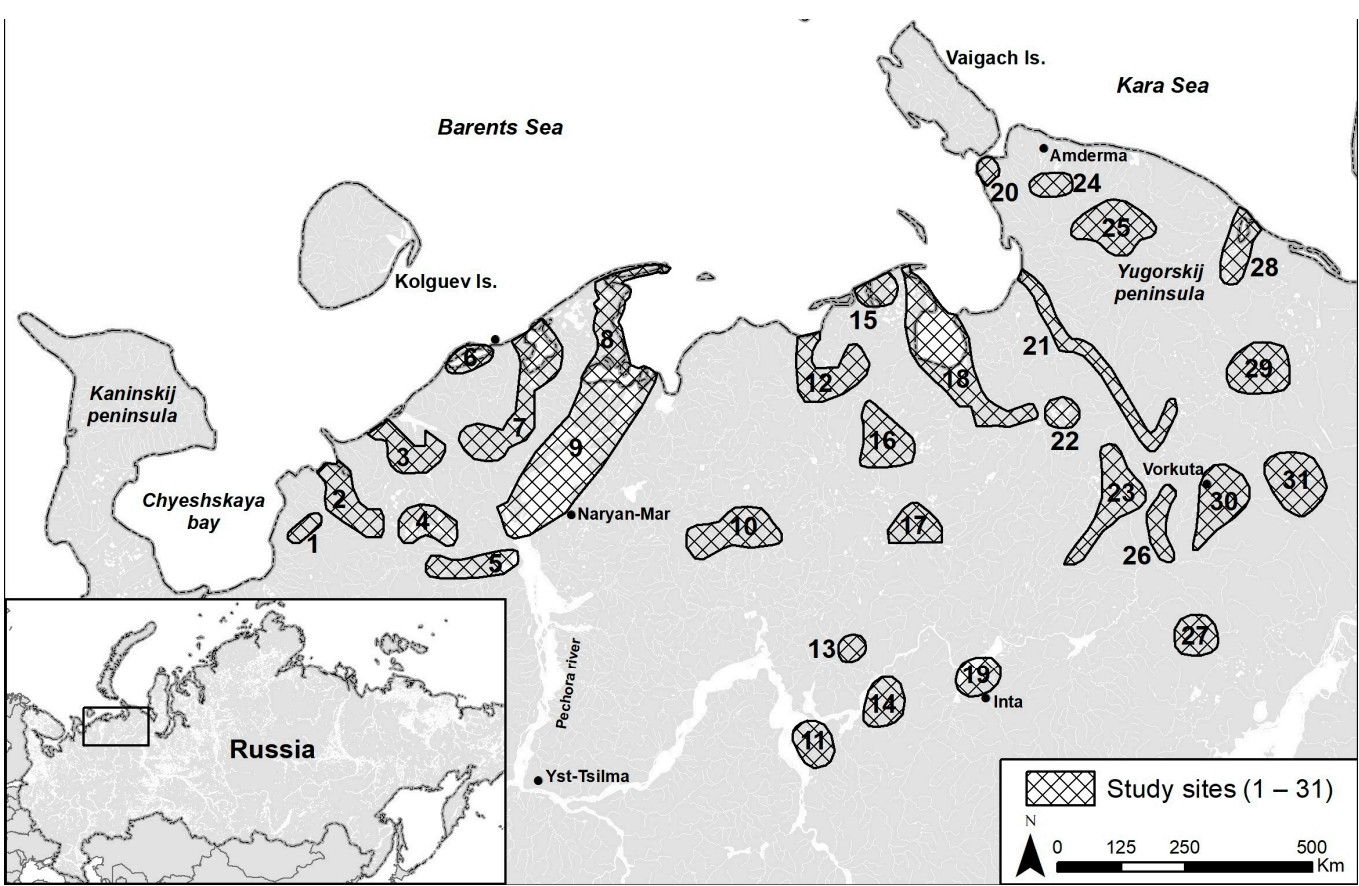

**Figure 1.** Sites where land-based and boat surveys of breeding Long-tailed Ducks were conducted.

The ANOVA test was used to estimate differences in the number, clutch, and brood sizes of Long-tailed Ducks. We also applied linear models to find the most significant factors affecting the Long-tailed Ducks. We considered the climatic data (annual temperature and precipitation) and the amount of oil production. Climate factors were calculated as average values from three weather stations (Indiga, Narjan-Mar, Amderma) on the sea coast of the study area. The data were taken from the Hydrometeorological database [22] and covered the entire period of duck observation. The amount of oil production in the region was considered the most significant anthropogenic factor influencing the number of birds.

We used the R (4.0.5) program [23] with the ggplot2 package [24] for statistical analysis and making diagrams. Google Earth 4.51 and GPS navigation were used to identify the geographic coordinates of survey areas. The Long-tailed Duck population density map was made by the interpolation method "Multilevel B-splines interpolation" in the program SAGA GIS (version 2.3.2).

## 3. Results

### 3.1. Breeding Ecology

A number of Long-tailed Ducks arrived at their breeding habitat already paired, while other birds formed pairs after arriving at the nesting grounds. The first eggs in nests were found in the MZT from 5 June to 10 July; in the BZT from 1 June to 22 June; and on the Yugorskij Peninsula from 19 June to 25 July. Up to 12 eggs were found in fully formed clutches, with an average clutch size of 5.9 eggs in the MZT (n = 37); 6.1 in the BZT (n = 165); and 5.6 on the Yugorskij Peninsula (n = 11). The earliest broods were observed from around 11 July in the BZT until 1 August on the Yugorskij Peninsula. The latest broods

were recorded on the19–20 August 2016. The average sizes of the broods ranged from 4.1 ducklings in the BZT (n = 45) to 6.1 in the MZT (n = 58). Certain broods, including those with 11–14 ducklings, probably belonged to two females.

The mean clutch and brood sizes did not differ significantly between the early study period of 1973–1997 and the late study period of 1998–2022 or between the BZT region and the MZT region (*p*-value for ANOVA: 0.883) (Table 1 and Figure 2).

**Table 1.** Two-way analysis of variance (ANOVA) for clutch and brood sizes between the years 1973–1997 vs. the years 1998–2022 (periods) and MZT vs. BZT (region).

| Effect | Df | Sum Sq | Mean Sq | F Value | Pr (>F) |
|---|---|---|---|---|---|
| | | | Clutch size | | |
| Period | 1 | 0.1 | 0.118 | 0.02 | 0.882 |
| Region | 1 | 17.8 | 17.78 | 3.34 | 0.070 |
| | | | Brood size | | |
| Period | 1 | 0.2 | 0.187 | 0.03 | 0.875 |
| Region | 1 | 8.4 | 8.406 | 1.12 | 0.293 |

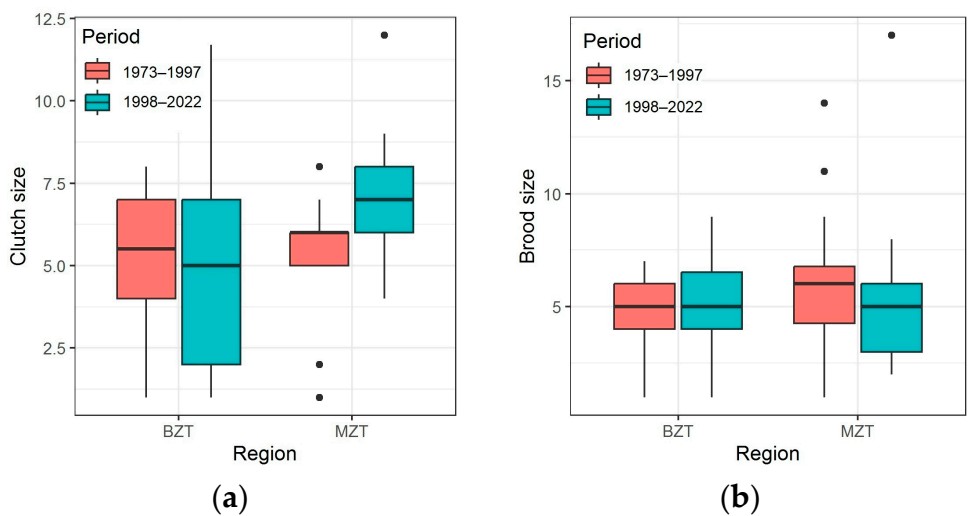

(**a**)　　　　　　　　　　　　　(**b**)

**Figure 2.** Boxplots of mean clutch (**a**) and brood (**b**) sizes of the Long-tailed Duck in the studied areas of the northeast European tundra from 1973–1997 and from 1998–2022. Dots mean outliers.

The two-way analysis of variance for clutch and brood sizes showed the absence of a statistically significant difference between the regions (the BZT and the MZT) and the periods (1973–1997 and 1998–2022) of observation (Table 1 and Figure 2).

### 3.2. Habitat Selection and Population Trends

In the northeast region of European Russia, Long-tailed Ducks breed in the tundra, the forest–tundra, and in the northernmost part of boreal forests. The species is most abundant in the tundra zone, where it can be found in almost all types of habitats, including in small tundra pools, lakes, rivers, and the coastal wetlands of the Barents and Kara Seas.

The breeding distribution range of the Long-tailed Duck stretches from coastal tundra habitats to lakes and marshes in boreal forests, with the southernmost breeding sites found up to 65°45′38″ N; 65°51′46″ E.

In different studied tundra areas, the breeding density of the Long-tailed Duck fluctuated from 0.6 up to 18.7 ind./km². The highest breeding densities were identified in 13 key breeding areas (Figure 3).

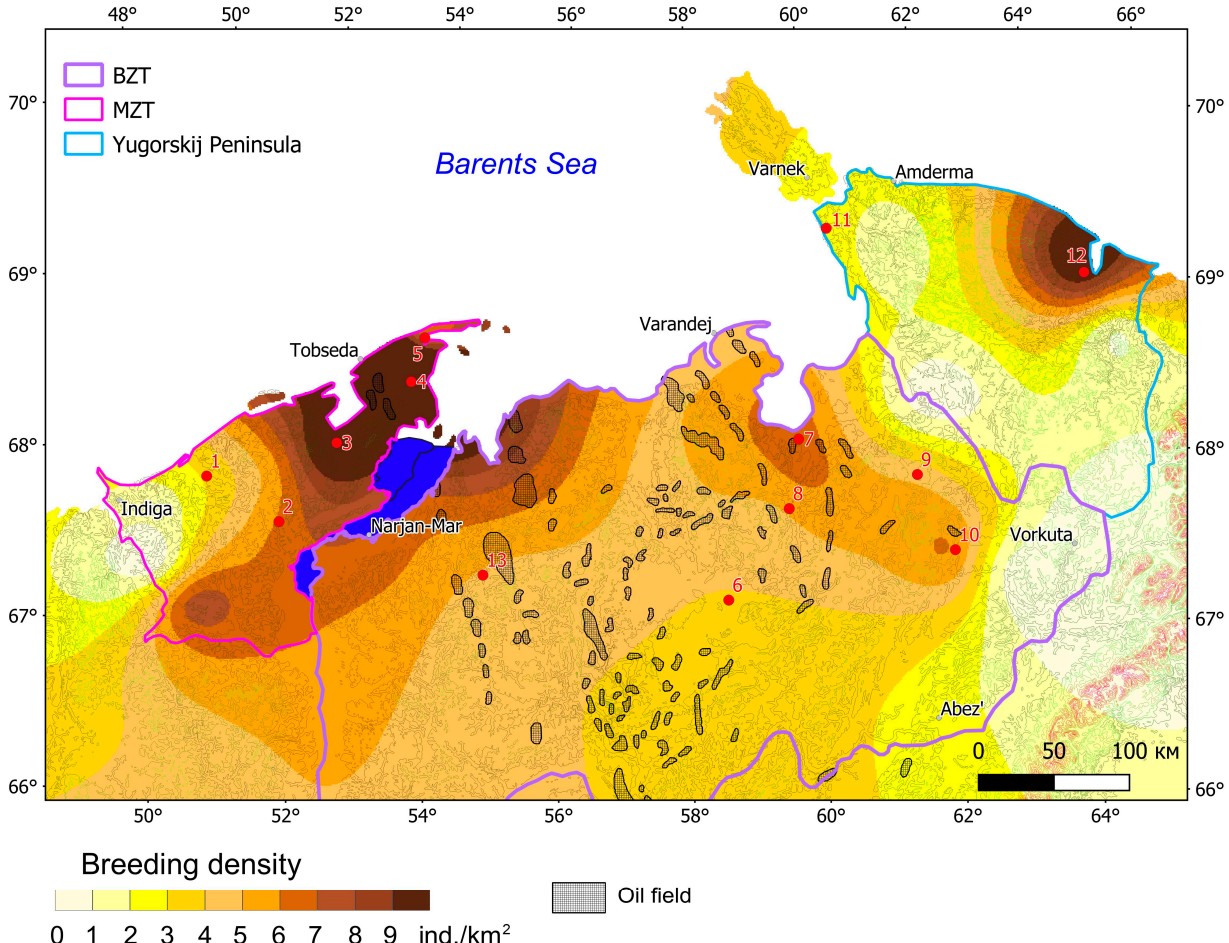

**Figure 3.** The breeding density of the Long-tailed Duck in the Northeast European tundra with the key breeding sites of the species (areas around red dots), along with the main gas and oil development sites. The sites were: 1—tundra at the lower Velt River (67°56′22.44′′ N; 50°20′47.53′′ E); 2—tundra at the lower Neruta River (68°12′21.95′′ N; 52°21′24.27′′ E); 3—Kolokolkova Bay area (68°28′44.43′′ N; 52°37′38.42′′ E); 4—Russkij Zavorot Peninsula (68°35′47.09′′ N; 53°29′31.20′′ E); 5—Kuznetskaya Bay area (68°51′38′′ N; 53°40′24′′ E); 6—Kolva River area (67°34′16.69′′ N;58°27′36.35′′ E); 7—Khaipudirskaya Bay area (68°28′6.54′′ N; 59°35′18.54′′ E); 8—the More-Y River area (68°6′52.79′′ N; 59°56′47.22′′ E); 9—Vashutkini lakes (68°1′7.83′′ N; 61°36′53.41′′ E); 10—the Bolshaya Rogovaya River basin (67°33′25.65′′ N;62°7′2.61′′ E); 11—Lymabadayaha-Sirtiyaha Rivers area (69°30′36.42′′ N; 60°23′5.13′′ E); 12—Kara Bay area (69°10′33.04′′; N; 64°48′4.69′′ E); 13—Shapkina River basin (67°28′16.79′′ N; 54°46′11.60′′ E).

The highest breeding density of the Long-tailed Duck in the entire northeast European tundra region was found in key areas located in the MZT (with up to 18.7 ind./km$^2$ recorded on the Russkij Zavorot Peninsula and up to 11.4 ind./km$^2$ in the Kolokolkova Bay). In the BZT, the largest breeding densities (up to 5.0 ind./km$^2$) were registered in the Shapkina River basin; in the Kolva River area (up to 5.0 ind./km$^2$); in the Khaipudirskaya Bay area (up to 6.4 ind./km$^2$); in the More-Y River area (up to 6.3 ind./km$^2$); at the Vashutkini Lakes (up to 4.7 ind./km$^2$); and in the Bolshaya Rogovaya River area (up to 5.9 ind./km$^2$). On the Yugorskij Peninsula, the most important breeding areas of the Long-tailed Duck were designated in the Kara Bay area, with up to 10.1 ind./km$^2$ recorded.

Large annual fluctuations in the breeding density of the Long-tailed Duck were characteristic in all studied sites. For example, in the key breeding area of the MZT (the Russkij Zavorot Peninsula), the density fluctuated from 3.5 ind./km$^2$ up to18.7 ind./km$^2$ during seventeen years of study. In the entire studied area of the MZT, the breeding density of

the species from 1986–2022 fluctuated from 0.6 ind./km$^2$ to18.7 ind./km$^2$, with an average calculated density of 6.5 ind./km$^2$ (Figure 4).

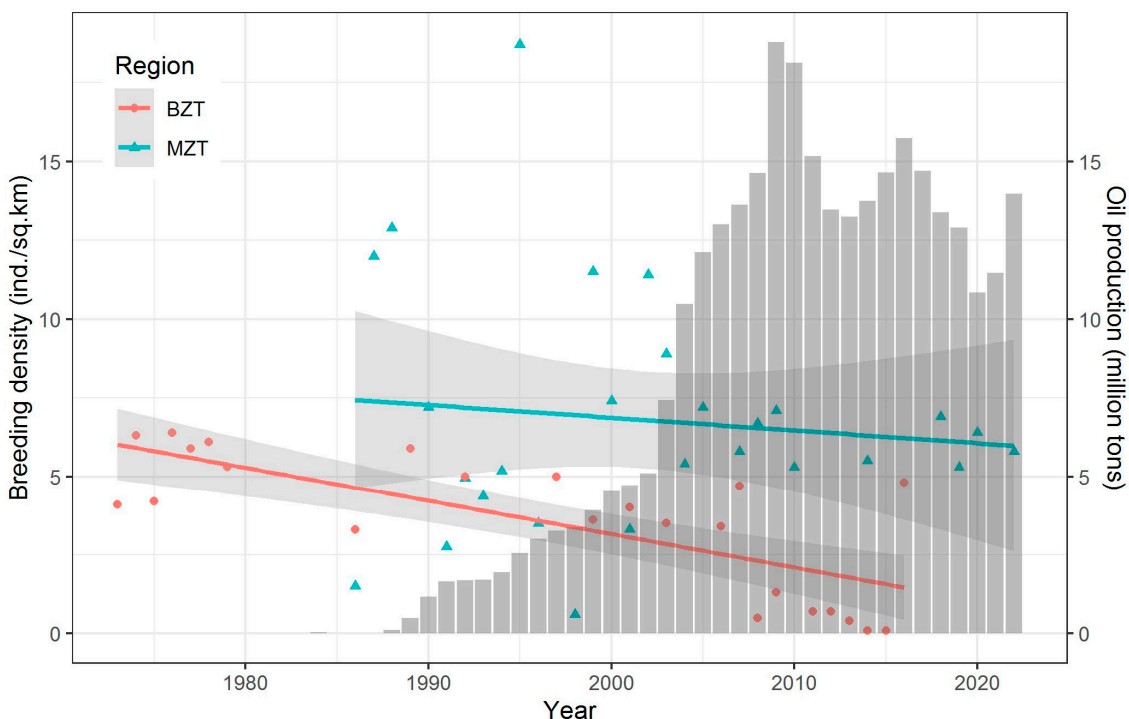

**Figure 4.** Estimated breeding densities of the Long-tailed Duck in the studied areas (lines and dots) and oil extraction in the Nenets Autonomous District (Russia) from 1984–2022 (columns).

According to the dispersion analysis, there was a statistically significant decrease in the number of Long-tailed Ducks in the BZT, not in the MZT, where the number of birds remained relatively stable (Table 2).

**Table 2.** Indicators of linear dependence models of the Long-tailed Duck population density of the observation years.

| Effect | Slope | t-Value | $p$ | DF | F | R$^2$ |
|---|---|---|---|---|---|---|
| | | | MZT | | | |
| Year | −0.040 | 0.560 | 0.580 | 1 | 0.31 | 0.01 |
| | | | BZT | | | |
| Year | **−0.106** | **5.44** | **<0.001** | 1 | 28.62 | 0.57 |

Statistically significant dependencies are in **bold**.

On the Yugorskij Peninsula, breeding densities recorded during eight years of study fluctuated from 0.6 ind./km$^2$ to10.1 ind./km$^2$, with an average calculated density of 3.1 ind./km$^2$.

In the studied area of the BZT, the breeding density of the species during 24 years of study fluctuated from 0.1 ind./km$^2$ to 6.4 ind./km$^2$, with an average calculated density of 3.6 ind./km$^2$.

A marked decline in breeding density during the last decade was recorded in most study areas of the BZT (Figure 4, Tables 2 and 3). For example, the registered breeding density in the Bolshaya Rogovaya River territory declined from 5.9 ind./km$^2$ in the 1980s to an estimated 3.3 ind./km$^2$ in the 2010s. In the Khaipudirskaya Bay area, the registered breeding density declined from 6.4 ind./km$^2$ in the 1970s to an estimated 3.4 ind./km$^2$ in the middle of the 2010s. In the Kolva River basin, the breeding density of ducks declined

from 5.0 ind./km$^2$ in 1990s to 3.4 ind./km$^2$ in the 2000s. In the More-Y River basin, the registered breeding density declined from 6.4 ind./km$^2$ in the 1970s to an estimated 3.1 ind./km$^2$ in the 2010s.

**Table 3.** Indicators of linear dependence models of the Long-tailed Duck population density depending on climatic parameters and oil production volume.

| Effect | Slope | t-Value | $p$ | F | R$^2$ |
|---|---|---|---|---|---|
| | | MZT | | | |
| (Intercept) | 4.865 | 0.506 | 0.618 | | |
| Temperature | 0.359 | 0.645 | 0.525 | 0.597 | 0.07 |
| Precipitation | 0.009 | 0.509 | 0.616 | | |
| Oil production | −0.0001 | 1.041 | 0.309 | | |
| | | BZT | | | |
| (Intercept) | **8.686** | **3.199** | **0.005** | | |
| Temperature | 0.274 | 1.127 | 0.273 | 11.08 | 0.62 |
| Precipitation | −0.005 | 1.011 | 0.324 | | |
| Oil production | **−0.0003** | **4.969** | **<0.001** | | |

Statistically significant dependencies are in **bold**.

Table 3 demonstrates the dependence of the Long-tailed Duck population density on climatic factors (average annual temperature and annual precipitation). The most significant factor was the volume of produced oil indicated for the BZT region. The rest of the factors turned out to be insignificant.

## 4. Discussion

More than 90% of all Long-tailed Ducks of the Western Siberia/Northern Europe population are found in the tundra zone of Russia [2,5]. The breeding population in European Russia was estimated to be about 5 million birds during the 1960s [25]. In the 1990s, the number declined to about 2 million birds, while the breeding population on the Yamal, Gydan, and Taymyr Peninsulas was estimated to be about 3.7 million individuals [26]. The total estimate of about 4.5 million birds breeding in the Arctic regions of European Russia and Western Siberia during that period was similar to that of the entire Western Siberia/Northern Europe population, estimated to be 4.6 million birds [27]. A marked decline in the number of breeding Long-tailed Ducks in northeast European Russia (except on the Kolguev and Vaigach Islands) was recorded during the first decade of the 21st century, with the total population estimated to be anywhere from 1.2–1.6 million individuals [17,18]. This number was similar to the estimated 1.5 million wintering birds counted in the Baltic Sea from 2007–2009 [4] and an overall estimate for the Western Siberia/Northern Europe population of 1.6 million birds [5]. A population decline in northeast European Russia also continued in the 2010s. We estimated the current breeding population of the Long-tailed Duck in European Russia to be about 0.9–1 million individuals.

Our data indicate that the loss or deterioration of key breeding habitats in the Arctic regions of Russia is possibly the main factor influencing the rapid population decline of the Long-tailed Duck. New oil and gas developments also negatively affect the key breeding sites of the species in Arctic Russia [28]. Oil extraction was authorized in the tundra zone of northeast European Russia in the 1990s, with more than 260,000,000 tons of oil extracted at the beginning of 2019. About 15 million tons of oil are extracted annually. There are 26 oil and gas extraction companies operating in the concerned region. A significant increase in oil extraction was registered in the region during the last two decades [29–34] (Figure 4).

In the BZT, the main sites of new oil and gas developments are located in the most important breeding areas for the Long-tailed Duck (Figure 3). Many key breeding habi-

tats were completely lost when they were transformed into major oil and gas extraction sites. The infrastructure of oil and gas developments included new roads and pipelines, the transportation of oil products, and the establishment of new settlements and towns. Therefore, the transformation of these areas led to increased disturbance and the large-scale deterioration of adjacent habitats, followed by the direct loss of certain key Long-tailed Duck nesting sites in the BZT. New oil and gas developments are associated with increased oil pollution and accidental oil spills. For example, during the field expedition in summer 2006 in the Ureryakha-Chernaya River basin, we crossed the territory of a large (100 sq. km plus the sanitary protection zone of 96 sq. km) operating oil field situated 200 km away from Naryan-Mar City. (We went down the river to our boats.) The river banks (at the segment about 10 km long) had traces of repeated oil spills, probably due to an accident of an oil pipeline crossing the river. In the 1980s, the study area served as an important breeding area for waterfowl and was included in the Perspective List of Ramsar Wetlands of Russia [35]. Unfortunately, at present, aquatic habitats (the Ureryakha and Chernaya Rivers) are heavily polluted with shale waters and oil products. Technogenic transformations of wetlands led to a decrease in the number of birds, primarily of the lake–river complex species [36].

Increased human disturbance is also an important factor negatively affecting Long-tailed Duck breeding. There was a statistically significant decrease in the Long-tailed Duck number in the BZT, where oil extraction largely increased. For the MZT, with no oil extraction, we observed no decrease in the number of birds (Figure 4 and Tables 2 and 3). Therefore, we supposed that oil extraction directly affected the number of birds. The increasing amount of extracted oil indirectly indicated that the volume of oil transported by tankers by sea is also increasing. This suggests that, along with the increase in the volume of oil transportation by sea, the probability of oil spills in the sea area will increase in direct proportion (during tanker refueling, accidents, etc.). Unfortunately, we do not have official data on oil spills in the Barents Sea, due to the unavailability of this information. The wing molt of the Long-tailed Duck occurs from late July to August in the Barents Sea. During this period, the flightless birds are particularly vulnerable to oil-spills. The most numerous groups of birds were recorded in the Khaipudyrskaya and Pakhancheskaya Bays, in shallow waters of the coast of the Medynskij Zavorot Peninsula, near the mouths of the Dresvyanka and Talotayakha Rivers, near the Belkovsky Nos Cape, and in the Yugorskij Shar Strait, as well as in the Kara Sea [17,18]. The Khaipudyrskaya and Pakhancheskaya Bays, the Dresvyanka area, and the water area close to the Medynsky Zavorot Peninsula have a very hard traffic of oil transportation by tankers. Flocks of molting ducks are at a high risk of being trapped in oil spills. For example, more than 2000 Long-tailed Ducks found on the sea beach were killed by an oil spill in the Khaipudyrskaya Bay of the Barents Sea in August 2003. However, even during seasonal (spring–autumn) migrations, this danger does not decrease, as oil and gas are transported by sea all year round, and ducks making stops for rest and food also have a chance of getting into oil spills.

The results of our research indicated a marked and rapid decline in the breeding population of the Long-tailed Duck in the BZT. Meanwhile, in the MZT, which was not affected by oil and gas development, the breeding population during the last decades only slightly decreased or remained stable in certain areas.

It is likely that the number of ducks killed annually in gillnets in Arctic Russia is high, but there are no reliable data about the effects of gillnet fishing on the local populations of Long-tailed Ducks. Similar to the situationin the Baltic Sea, Long-tailed Ducks are regularly killed by becoming entangled in gillnets used for inshore fishing in the Barents and Kara Seas, as well as in inland tundra lakes. There are episodic cases of fixing the deaths of Long-tailed Ducks in gillnets. During our expedition work on June 18 of 1989 on Lake Gitara in the basin of the Bolshaya Rogovaya River (the BZT), 11 Long-tailed Ducks (10 males and 1 female) got into the poachers' net (70 m long). On June 19 of 2006, in the area of a large operating oil field on the Ureryakha River (the BZT) on the Urerkhasyrey Lake, we recorded a placed net of 150 m long. A total of 20 Long-tailed Ducks got into the net; most of them drowned by the time of registration. In addition to what has been said

about the impact of oil and gas extraction in the BZT and anthropogenic transformation, it must be said that these nets were placed by staff of the oil field. Fishing was carried out illegally. Driving an all-terrain vehicle across the tundra from the oil field to the lake was also illegal. Unfortunately, such a picture is not uncommon, especially at oil (gas) fields remote from the center (Naryan-Mar City), as inspections are not carried out there or are carried out very rarely.

The extreme change in the benthic macrofauna community after the invasion of the bottom fish Round Goby (*Neogobius melanostomus*) in the eastern Baltic Sea changed the diet of the Long-tailed Duck at the wintering grounds. However, the Long-tailed Duck switched to another food supply. This indicates a high adaptability of the species [37].

As a result of our study, it was shown that temperature and precipitation changes did not significantly affect the decrease in the number of Long-tailed Ducks.

The cumulative negative impacts of multiple anthropogenic factors on the breeding population of the Long-tailed Duck in the tundra zone of northeast European Russia are significant, particularly in view of possible increases in oil and gas developments, along with a corresponding increase in marine traffic.

The results of our study highlight the importance of conservation measures, including the establishment of new protections in the key breeding areas of the Long-tailed Duck and in particular, the MZT. The enhancement of research programs for this species in its principal European breeding habitats is also warranted.

## 5. Conclusions

Our study showed that the major decline in the Long-tailed Duck population in northeast European Russia, from about 5 million birds in the 1960s to 1 million individuals today, was mainly caused by the loss or deterioration of the key breeding habitats of the species resulting from new oil and gas developments, as well as oil and gas transportation by sea. Oil pollution from shipping and gillnet fishing bycatching of this vulnerable species in the Baltic Sea were also important factors influencing the population decline. We found that the temperature and precipitation changes did not significantly affect the size of the Long-tailed Duck population. The Long-tailed Duck is a highly adaptable species, and even changes in the food resources at the wintering grounds did not significantly affect its population density.

**Author Contributions:** Conceptualization, methodology, investigation, writing—original draft preparation, and data curation: O.M., Y.M., and S.K.; software and statistical data analysis: A.N.; writing—review and editing: O.M. All authors have read and agreed to the published version of the manuscript.

**Funding:** The study was carried out within the state assignment No 122040600025-2 at the Institute of Biology, Komi Scientific Centre of the Ural Branch of the Russian Academy of Sciences.

**Institutional Review Board Statement:** Field surveys of Long-tailed Ducks were conducted following the research ethic rules of the Institute of Biology, ensuring minimal disturbance to the breeding birds.

**Data Availability Statement:** It is possible to submit the manuscript data for open access publication under the terms and conditions of the Creative Commons Attribution.

**Acknowledgments:** The authors are deeply grateful to Saulius Švažas for his help in preparing the article at all stages. The authors thank Yulia Leonova and Yulia Bogomolova from the Nenetsky State Reserve, who provided information on the number of Long-tailed Duck is the territory of the Nenetsky State Reserve in certain years. We thank Vladimir Shchanov for his help in the preparation of the Long-tailed Duck population density map.

**Conflicts of Interest:** The authors declare no conflict of interest.

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
