# Peer review of "Population Status of the Globally Threatened Long-Tailed Duck Clangula hyemalis in the Northeast European Tundra"

_diversity, doi:10.3390/d15050666_

Round 1
Reviewer 1 Report
Comments on the MS Population Status of the Globally Threatened Long-tailed Duck Clangula hyemalis in the North-East European Tundra by Mineev et al.
In the MS Mineev et al. describe the density of breeding of long-tailed ducks at different sites within the European part of the Russian arctic tundra. They also compare mean clutch and brood sizes between two regions and between two time periods. Yearly values of breeding densities within two regions are also presented. They conclude that habitat destruction by oil and gas production has caused the decrease in breeding density in at least one important region.
Below follow specific comments on the MS:
Line 19. The authors conclude that
“Our data indicate that the loss or deterioration of key breeding habitats in the Arctic regions of Russia is the main factor influencing rapid population decline.”
Mineev et al. describe a loss of breeding habitats within one important region, but what about the deterioration of other regions. Perhaps the words “is the main factor” should be changed to “is one important factor”.
Check that the final wording also correspond to what is written in the conclusion
Line 40. The sentence “The majority of the global population overwinters in the Baltic Sea” should be clarified.
Perhaps to “The majority of the birds belonging to the North European / West Siberian population overwinters in the Baltic Sea”
Line 55. The authors could also clarify for the reader that a proportion of the birds that breed more eastward (in Asia) between Yugorski peninsula and Taymyr peninsula also may winter in the Baltic Sea (if that is true).
Figure 1 and line 73. The borders of the five major regions should also be indicated in the map.
Line 114-122. The sentence “A certain stable state in clutch and brood sizes were found” is unclear.
I guess that one can write “The mean clutch and brood sizes did not differ significantly between the early study period 1973-1997 and the late study period 1998-2022. Nor did the mean clutch and brood sizes differ between the BZT region and the MZT region.
Figure 2 legend. Clarify what the boxes and error bars show. Is it standard deviations and range? What does dots indicate? Outliers?
Figure 3 and legend. Describe what the different colours indicate. Which densities are represented by which colours. Clarify if the grey small areas represent gas and oil development sites.
Line 152 The highest breeding density ….
Line 152-162. The density data given here could instead be given in a table.
Line 163. The annual fluctuations are interesting and could be more thoroughly discussed.
Line 190-198. The data here could be given in a figure. It would be interesting to see if annual fluctuations are correlated among sites.
Figure 4. Secondary Y-axis title must be wrong (should be tons). I guess that the area is not producing up to 15 million tons per year. Breeding density - give units ind /km2 at axis-title
Clarify if the huge annual variation in breeding density in the MZT region before 2003 is considered real or perhaps due to small sample sizes. If real – why so large annual variation before 2003 and small variation later on?
Are there any explanation for the relatively high density in the summer 2016 in the BZT region? If the low values between 2008 and 2015 are due the habitat destruction, how come the high value in 2016? Clarify if possible.
Table 2. If this is the result of a linear regression then DF should be 1.
Table 3. Why not a t-test, a comparison of mean values between two periods. This table could be deleted. Table 2 and Figure 4 seems enough.
Line 187 23 or 24 years?
Line 210-216. These figures do not match. If numbers in summer in north European Russia was 1.2-1.6 million birds and the number of wintering birds in the Baltic Sea was 1.5 million than the population between Yamal and Taymyr must have become very small. Is that realistic?
Author Response
Dear Reviewer! Tank you for your comments and corrections, they were very valuable! We have prepared answers on your points. Some of them are not in the borders of the manuscript, but of some strong research. I hope we will meet, and discus all of questions

Reviewer 2 Report
This paper assesses the distribution and abundance of Long-tailed ducks breeding in north-east European tundra of Russia. The authors use data from boat and land surveys roughly from 1973 to 2023 in key breeding areas to assess population change. They also report data on arrival/departure dates, clutch and brood sizes from a few breeding sites within the survey areas. The premise of their findings is that oil and gas developments have caused significant breeding habitat loss and, thereby, a decrease in numbers of breeding long-tailed ducks in the area. Overall with some heavy revisions, the manuscript will provide a comprehensive assessment of this critical breeding habitat since the 1970s. This manuscript has merit for publication, but needs more work to make it a stronger paper and to enhance the author's conclusions.

Round 2
Reviewer 2 Report
I think the flow and focus of the paper have improved tremendously. The additional analyses help clarify that you consider environmental variability as a factor influencing LTDU productivity and habitat use. I caution stating the climate change had no significant influence on LTDUs in the area. Looking at ambient temperature and precipitation is not sufficient to state that these factors represent climate change. That is a big leap. Just stick to what you tested...that temperature and precipitation did not impact LTDU habitat use.
Author Response
I agree with your remark. I've generalized too much. Corrected.